# Nutrition, Food and Diet in Health and Longevity: We Eat What We Are

**DOI:** 10.3390/nu14245376

**Published:** 2022-12-18

**Authors:** Suresh I. S. Rattan, Gurcharan Kaur

**Affiliations:** 1Department of Molecular Biology and Genetics, Aarhus University, 8000 Aarhus, Denmark; 2Department of Biotechnology, Guru Nanak Dev University, Amritsar 143005, India

**Keywords:** healthy ageing, macro- and micro-nutrients, diet and culture, nutritional interventions

## Abstract

Nutrition generally refers to the macro- and micro-nutrients essential for survival, but we do not simply eat nutrition. Instead, we eat animal- and plant-based foods without always being conscious of its nutritional value. Furthermore, various cultural factors influence and shape our taste, preferences, taboos and practices towards preparing and consuming food as a meal and diet. Biogerontological understanding of ageing has identified food as one of the three foundational pillars of health and survival. Here we address the issues of nutrition, food and diet by analyzing the biological importance of macro- and micro-nutrients including hormetins, discussing the health claims for various types of food, and by reviewing the general principles of healthy dietary patterns, including meal timing, caloric restriction, and intermittent fasting. We also present our views about the need for refining our approaches and strategies for future research on nutrition, food and diet by incorporating the molecular, physiological, cultural and personal aspects of this crucial pillar of health, healthy ageing and longevity.

## 1. Introduction

The terms nutrition, food and diet are often used interchangeably. However, whereas nutrition generally refers to the macro- and micro-nutrients essential for survival, we do not simply eat nutrition, which could, in principle, be done in the form of a pill. Instead, we eat food which normally originates from animal- and plant-based sources, without us being aware of or conscious of its nutritional value. Even more importantly, various cultural factors influence and shape our taste, preferences, taboos and practices towards preparing and consuming food as a meal and diet [1]. Furthermore, geo-political-economic factors, such as governmental policies that oversee the production and consumption of genetically modified foods, geological/climatic challenges of growing such crops in different countries, and the economic affordability of different populations for such foods, also influence dietary habits and practices [2,3]. On top of all this lurks the social evolutionary history of our species, previously moving towards agriculture-based societies from the hunter-gatherer lifestyle, now becoming the consumers of industrially processed food products that affect our general state of health, the emergence of diseases, and overall lifespan [1,4]. The aim of this article is to provide a commentary and perspective on nutrition, food and diet in the context of health, healthy ageing and longevity.

Biogerontological understanding of ageing has identified food as one of the three foundational pillars of health and survival. The other two pillars, especially in the case of human beings, are physical exercise and socio-mental engagement [5,6,7]. A huge body of scientific and evidence-based information has been amassed with respect to the qualitative and quantitative nature of optimal nutrition for human health and survival. Furthermore, a lot more knowledge has developed regarding how different types of foods provide different kinds of nutrition to different extents, and how different dietary practices have either health-beneficial or health-harming effects.

Here we endeavor to address these issues of nutrition, food and diet by analyzing the biological importance of macro- and micro-nutrients, and by discussing the health-claims about animal-based versus plant-based foods, fermented foods, anti-inflammatory foods, functional foods, foods for brain health, and so on. Finally, we discuss the general principles of healthy dietary patterns, including the importance of circadian rhythms, meal timing, chronic caloric restriction (CR), and intermittent fasting for healthy ageing and extended lifespan [8,9]. We also present our views about the need for refining our approaches and strategies for future research on nutrition, food and diet by incorporating the molecular, physiological, cultural and personal aspects of this crucial pillar of health, healthy ageing and longevity.

## 2. Nutrition for Healthy Ageing

The science of nutrition or the “nutritional science” is a highly advanced field of study, and numerous excellent books, journals and other resources are available for fundamental information about all nutritional components [10]. Briefly, the three essential macronutrients which provide the basic materials for building biological structures and for producing energy required for all physiological and biochemical processes are proteins, carbohydrates and lipids. Additionally, about 18 micronutrients, comprised of minerals and vitamins, facilitate the optimal utilization of macronutrients via their role in the catalysis of numerous biochemical processes, in the enhancement of their bioavailability and absorption, and in the balancing of the microbiome. Scientific literature is full of information about almost all nutritional components with respect to their importance and role in basic metabolism for survival and health throughout one’s life [10].

In the context of ageing, a major challenge to maintain health in old age is the imbalanced nutritional intake resulting into nutritional deficiency or malnutrition [11,12]. Among the various reasons for such a condition is the age-related decline in the digestive and metabolic activities, exacerbated by a reduced sense of taste and smell and worsening oral health, including the ability to chew and swallow [13,14]. Furthermore, an increased dependency of the older persons on medications for the management or treatment of various chronic conditions can be antagonistic to certain essential nutrients. For example, long term use of metformin, which is the most frequently prescribed drug against Type 2 diabetes, reduces the levels of vitamin B12 and folate in the body [15,16]. Some other well-known examples of the drugs used for the management or treatment of age-related conditions are cholesterol-lowering medicine statin which can cause coenzyme Q10 levels to be too low; various diuretics (water pills) can cause potassium levels to be too low; and antacids can decrease the levels of vitamin B12, calcium, magnesium and other minerals [15,16]. Thus, medications used in the treatment of chronic diseases in old age can also be “nutrient wasting” or “anti-nutrient” and may cause a decrease in the absorption, bioavailability and utilization of essential micronutrients and may have deleterious effects to health [11]. In contrast, many nutritional components have the potential to interact with various drugs leading to reduced therapeutic efficacy of the drug or increased adverse effects of the drug, which can have serious health consequences. For example, calcium in dairy products like milk, cheese and yoghurt can inhibit the absorption of antibiotics in the tetracycline and quinolone class, thus compromising their ability to treat infection effectively. Some other well-known examples of food sources which can alter the pharmacokinetics and pharmacodynamics of various drugs are grape fruits, bananas, apple juice, orange juice, soybean flour, walnuts and high-fiber foods (see: https://www.aarp.org/health/drugs-supplements/info-2022/food-medication-interaction.html (accessed on 13 November 2022)).

It is also known that the nutritional requirements of older persons differ both qualitatively and quantitatively from young adults [11]. This is mainly attributed to the age-related decline in the bioavailability of nutrients, reduced appetite, also known as ‘anorexia of ageing,’ as well as energy expenditure [12,17,18]. Therefore, in order to maintain a healthy energy balance, the daily uptake of total calories may need to be curtailed without adversely affecting the nutritional balance. This may be achieved by using nutritional supplements with various vitamins, minerals and other micronutrients, without adding to the burden of total calories [12,17,18]. More recently, the science of nutrigenomics (how various nutrients affect gene expression), and the science of nutrigenetics (how individual genetic variations respond to different nutrients) are generating novel and important information on the role of nutrients in health, survival and longevity.

## 3. Food for Healthy Ageing

The concept of healthy ageing is still being debated among biogerontologists, social-gerontologists and medical practioners. It is generally agreed that an adequate physical and mental independence in the activities of daily living can be a pragmatic definition of health in old age [7]. Thus, healthy ageing can be understood as a state of maintaining, recovering and enhancing health in old age, and the foods and dietary practices which facilitate achieving this state can be termed as healthy foods and diets.

From this perspective, although nutritional requirements for a healthy and long life could be, in principle, fulfilled by simply taking macro- and micro-nutrients in their pure chemical forms, that is not realistic, practical, attractive or acceptable to most people. In practice, nutrition is obtained by consuming animals and plants as sources of proteins, carbohydrates, fats and micronutrients. There is a plethora of tested and reliable information available about various food sources with respect to the types and proportion of various nutrients present in them. However, there are still ongoing discussions and debates as to what food sources are best for human health and longevity [19,20]. Often such discussions are emotionally highly charged with arguments based on faith, traditions, economy and, more recently, on political views with respect to the present global climate crisis and sustainability.

Scientifically, there is no ideal food for health and longevity. Varying agricultural and food production practices affect the nutritional composition, durability and health beneficial values of various foods. Furthermore, the highly complex “science of cooking” [21], evolved globally during thousands of years of human cultural evolution, has discovered the pros and cons of food preparation methods such as soaking, boiling, frying, roasting, fermenting and other modes of extracting, all with respect to how best to use these food sources for increasing the digestibility and bioavailability of various nutrients, as well as how to eliminate the dangers and toxic effects of other chemicals present in the food.

The science of food preparation and utilization has also discovered some paradoxical uses of natural compounds, especially the phytochemicals such as polyphenols, flavonoids, terpenoids and others. Most of these compounds are produced by plants as toxins in response to various stresses, and as defenses against microbial infections [22,23]. However, humans have discovered, mostly by trial and error, that numerous such toxic compounds present in algae, fungi, herbs and other sources can be used in small doses as spices and condiments with potential benefits of food preservation, taste enhancement and health promotion [23].

The phenomenon of “physiological hormesis” [24] is a special example of the health beneficial effects of phytotoxins. According to the concept of hormesis, a deliberate and repeated use of low doses of natural or synthetic toxins in the food can induce one or more stress responses in cells and tissues, followed by the stimulation of numerous defensive repair and maintenance processes [25,26]. Such hormesis-inducing compounds and other conditions are known as hormetins, categorized as nutritional, physical, biological and mental hormetins [27,28,29]. Of these, nutritional hormetins, present naturally in the food or as synthetic hormetins to be used as food supplements, are attracting great attention from food-researchers and the nutraceutical and cosmeceutical industry [27,30]. Other food supplements being tested and promoted for health and longevity are various prebiotics and probiotics strengthening and balancing our gut microbiota [31,32,33].

Recently, food corporations in pursuit of both exploiting and creating a market for healthy ageing products, have taken many initiatives in producing new products under the flagship of nutraceuticals, super-foods, functional foods, etc. Such products are claimed and marketed not only for their nutritional value, but also for their therapeutic potentials [10]. Often the claims for such foods are hyped and endorsed as, for example, anti-inflammatory foods, food for the brain, food for physical endurance, complete foods, anti-ageing foods and so on [34,35,36]. Traditional foods enriched with a variety of minerals, vitamins and hormetins are generally promoted as “functional foods” [37]. Even in the case of milk and dairy products, novel and innovative formulations are claimed to improve their functionality and health promotional abilities [38]. However, there is yet a lot to be discovered and understood about such reformulated, fortified and redesigned foods with respect to their short- and long-term effects on physiology, microbiota balance and metabolic disorders in the context of health and longevity.

## 4. Diet and Culture for Healthy and Long Life

What elevates food to become diet and a meal is the manner and the context in which that food is consumed [4]. Numerous traditional and socio-cultural facets of dietary habits can be even more significant than their molecular, biochemical, and physiological concerns regarding their nutritional ingredients and composition. For example, various well-known diets, such as the paleo, the ketogenic, the Chinese, the Ayurvedic, the Mediterranean, the kosher, the halal, the vegetarian, and more recently, the vegan diet, are some of the diverse expressions of such cultural, social, and political practices [1]. The consequent health-related claims of such varied dietary patterns have influenced their acceptance and adaptation globally and cross-culturally.

Furthermore, our rapidly developing understanding about how biological daily rhythms affect and regulate nutritional needs, termed “chrono-nutrition”, has become a crucial aspect of optimal and healthy eating habits [39,40]. A similar situation is the so-called “nutrient timing” that involves consuming food at strategic times for achieving certain specific outcomes, such as weight reduction, muscle strength, and athletic performance. The meal-timing and dietary patterns are more anticipatory of health-related outcomes than any specific foods or nutrients by themselves [41,42,43,44]. However, encouraging people to adopt healthy dietary patterns and meal-timing requires both the availability, accessibility and affordability of food, and the intentional, cultural and behavioral preferences of the people.

Looking back at the widely varying and constantly changing cultural history of human dietary practices, one realizes that elaborate social practices, rituals and normative behaviors for obtaining, preparing and consuming food, are often more critical aspects of health-preservation and health-promotion than just the right combination of nutrients. Therefore, one cannot decide on a universal food composition and consumption pattern ignoring the history and the cultural practices and preferences of the consumers. After all, “we eat what we are”, and not, as the old adage says, “we are what we eat”.

## 5. Conclusions and Perspectives

Food is certainly one of the foundational pillars of good and sustained health. Directed and selective evolution through agricultural practices and experimental manipulation and modification of food components have been among the primary targets for improving food quality. This is further authenticated by extensive research performed, mainly on experimental animal and cell culture model systems, demonstrating the health-promoting effects of individual nutritional components and biological extracts in the regulation, inhibition or stimulation of different molecular pathways with reference to healthy ageing and longevity [45]. Similarly, individual nutrients or a combination of a few nutrients are being tested for their potential use as calorie restriction mimetics, hormetins and senolytics [46,47,48]. However, most commonly, these therapeutic strategies follow the traditional “one target, one missile” pharmaceutical-like approach, and consider ageing as a treatable disease. Based on the results obtained from such experimental studies, the claims and promises made which can often be either naïve extrapolations from experimental model systems to human applications, or exaggerated claims and even false promises [49].

Other innovative, and possibly holistic, food- and diet-based interventional strategies for healthy ageing are adopting regimens such as caloric- and dietary-restriction, as well as time-restricted eating (TRE). Intermittent fasting (IF), the regimen based on manipulating the eating/fasting timing, is another promising interventional strategy for healthy ageing. Chrono-nutrition, which denotes the link between circadian rhythms and nutrient-sensing pathways, is a novel concept illustrating how meal timings alignment with the inherent molecular clocks of the cells functions to preserve metabolic health. TRE, which is a variant of the IF regimen, claims that food intake timing in alignment with the circadian rhythm is more beneficial for health and longevity [39,40,41,50]. Moreover, TRE has translational benefits and is easy to complete in the long term as it only requires limiting the eating time to 8–10 h during the day and the fasting window of 12–16 h without restricting the amount of calories consumed. Some pilot studies on the TRE regimen have reported improvement in glucose tolerance and the management of body weight and blood pressure in obese adults as well as men at risk of T2D. Meta-analyses of several pilot scale studies in human subjects suggest and support the beneficial effects of a TRE regimen on several health indicators [39,50]. Several other practical recommendations, based on human clinical trials have also been recommended for meeting the optimal requirements of nutrition in old age, and for preventing or slowing down the progression of metabolic syndromes [39,40,41,50].

What we have earlier discussed in detail [4] is supported by the following quote: “…food is more than just being one of the three pillars of health. Food is both the foundation and the scaffolding for the building and survival of an organism on a daily basis. Scientific research on the macro- and micro-nutrient components of food has developed deep understanding of their molecular, biochemical and physiological roles and modes of action. Various recommendations are repeatedly made and modified for some optimal daily requirements of nutrients for maintaining and enhancing health, and for the prevention and treatment of diseases. Can we envisage developing a “nutrition pill” for perfect health, which could be used globally, across cultures, and at all ages? We don’t think so” [4].

Our present knowledge about the need and significance of nutrients is mostly gathered from the experimental studies using individual active components isolated from various food sources. In reality, however, these nutritional components co-exist interactively with numerous other compounds, and often become chemically modified through the process of cooking and preservation, affecting their stability and bioavailability. There is still a lot to be understood about how the combination of foods, cooking methods and dietary practices affect health-related outcomes, especially with respect to ageing and healthspan.

An abundance of folk knowledge in all cultures about food-related ‘dos and don’ts’ requires scientific verification and validation. We also need to reconsider and change our present scientific protocols for nutritional research, which seem to be impractical for food and dietary research at the level of the population. It is a great scientific achievement that we have amassed a body of information with respect to the nature of nutritional components required for health and survival, the foods which can provide those nutritional components and the variety of dietary and eating practices which seem to be optimal for healthy survival and longevity.

Finally, whereas abundant availability of and accessibility to food in some parts of the world has led to over-consumption and consequent life-style-induced metabolic diseases and obesity, in many other parts of the world food scarcity and economic disparity continue to perpetuate starvation, malnutrition, poor health and shortened lifespan. Often, it is not a lack of knowledge about the optimal nutrition, food and diet that leads to making bad choices; rather, it is either our inability to access and afford healthy foods or our gullibility to fall prey to the exaggerated claims in the commercial interests of food producing and marketing companies. We must continue to gather more scientific information and knowledge about the biochemical, physiological and cultural aspects of nutrition, food and diet, which should then be recommended and applied wisely and globally, incorporating the social, cultural and environmental needs of all. After all, “we eat what we are”, and not merely “we are what we eat”!

## Data Availability

Not applicable.

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
