# Peer review of "Nutrition, Food and Diet in Health and Longevity: We Eat What We Are"

_nutrients, 2022, doi:10.3390/nu14245376_

Round 1
Reviewer 1 Report
This article briefly reviews the associations between nutrition, food, diet, and healthy aging/longevity. Overall, the paper is easy to understand and follow. The structure is also clear. The authors' proposed call for refining our approaches and strategies for future research on nutrition, food, and diet via incorporating the molecular, physiological, cultural, and personal aspects is important and should be taken as a guideline for future research.
I enjoyed reading it. I have some mild comments to share with the authors.
1. Not sure the first two sentences in the first paragraph of the Introduction are needed (this is also relevant to the first sentence in the Abstract.
2. Lines 31-32: The author mentioned geo-political factors. It is unclear to me how political factors are associated with diet. More elucidation is needed.
3. Healthy diet and healthy foods or healthy eating are common jargon terms in the field. The authors may consider reviewing them or expanding the current paragraphs to include their definitions and more recent research findings.
4. The author mentioned intervention programs, but this part is weak and needs to be strengthened.
5. Healthy aging seems not clearly defined in the article.
6. It would be super to provide some empirical data in commenting/reviewing/arguing some points to make arguments more strong and more concrete.
Reviewer 2 Report
This is a good commentary about nutrition, food, and diet in the context of health, healthy aging, and longevity.
